

# Sensitivity Analysis of the Surface Ozone and Fine
# Particulate Matter to Meteorological Parameters in China
Zhihao Shi[1], Lin Huang[1], Jingyi Li[1], Qi Ying[2], Hongliang Zhang[3,4], Jianlin Hu[1*]
[1]Jiangsu Key Laboratory of Atmospheric Environment Monitoring and Pollution
Control, Collaborative Innovation Center of Atmospheric Environment and
Equipment Technology, Nanjing University of Information Science & Technology,
Nanjing 210044, China
[2]Zachry Department of Civil and Environmental Engineering, Texas A&M University,
College Station, TX 77843, USA
[3]Department of Environmental Science and Engineering, Fudan University, Shanghai
200438, China
[4]Institute of Eco-Chongming (SIEC), Shanghai 200062, China
[*]Corresponding authors:
Jianlin Hu, Email: jianlinhu@nuist.edu.cn. Phone: +86-25-58731504.



## Abstract

Meteorological conditions play important roles in the formation of ozone ($O_3$) and fine particulate matter ($PM_{2.5}$). China has been suffering from serious regional air pollution problems, characterized by high concentrations of surface $O_3$ and $PM_{2.5}$. In this study, the Community Multiscale Air Quality (CMAQ) model was used to quantify the sensitivity of surface $O_3$ and $PM_{2.5}$ to key meteorological parameters in different regions of China. Six meteorological parameters were perturbed to create different meteorological conditions, including temperature (T), wind speed (WS), absolute humidity (AH), planetary boundary layer height (PBLH), cloud liquid water content (CLW) and precipitation (PCP). Air quality simulations under the perturbed meteorological conditions were conducted in China in January and July of 2013. The changes in $O_3$ and $PM_{2.5}$ concentrations due to individual meteorological parameters were then quantified. T has the greatest impact on the daily maximum 8-h average $O_3$ ($O_3$-8h) concentrations, which leads to $O_3$-8h increases by 1.7 ppb $K^{-1}$ in January in Chongqing and 1.1 ppb $K^{-1}$ in July in Beijing. WS, AH, and PBLH have a smaller but notable influence on $O_3$-8h with maximum change rates of 0.3, -0.15, and 0.14 ppb $\%^{-1}$, respectively. T, WS, AH, and PBLH have important effects on $PM_{2.5}$formation of in both January and July. In general, $PM_{2.5}$ sensitivities are negative to T, WS, and PBLH and positive to AH in most regions of China. The sensitivities in January are much larger than in July. $PM_{2.5}$ sensitivity to T, WS, PBLH, and AH in January can be up to -5 μg $m^{-3}$ $K^{-1}$ , -3 μg $m^{-3}$ $\%^{-1}$ , -1 μg $m^{-3}$ , and +0.6 μg $m^{-3}$ $\%^{-1}$, respectively, and in July can be up to -2 μg $m^{-3}$ $K^{-1}$, -0.4 μg $m^{-3}$ $\%^{-1}$, -0.14 μg $m^{-3}$ $\%^{-1}$, and +0.3 μg $m^{-3}$ $\%^{-1}$, respectively. Other meteorological factors (CLW and PCP) have negligible effects on $O_3$-8h (less than 0.01 ppb $\%^{-1}$) and $PM_{2.5}$ (less than 0.01 μg $m^{-3}$ $\%^{-1}$). The results suggest that surface $O_3$ and $PM_{2.5}$ concentrations can



change significantly due to changes in meteorological parameters and it is necessary
to consider these effects when developing emission control strategies in different
regions of China.

Keywords: sensitivity, meteorological conditions, fine particulate matter, ozone,
CMAQ model


# 1. **Introduction**

China has serious air pollution problems and fine particulate matter ($PM_{2.5}$) and ozone ($O_3$) are the two major air pollutants (Lin et al., 2010; Hu et al., 2016; Lu et al., 2019; Wu et al., 2019). The annual average $PM_{2.5}$ concentrations were higher than 50 µg m$^{-3}$ in 26 out of the total 31 provincial capital cities in mainland China during 2013-2014 (Wang et al., 2014a), and the national 4[th] highest daily maximum 8-hour average $O_3$ ($O_3$-8h) is 86.0 ppb during the warm-seasons (April–September) in 2013-2017, which is 6.3–30% higher than that in other industrialized regions of the world (Lu et al., 2018). $PM_{2.5}$ alone caused 0.87-1.36 million deaths every year in China, and long-term exposure to $O_3$ was responsible for an extra 254 000 deaths (Apte et al., 2015; Cohen et al., 2017; Hu et al., 2017b; Silver et al., 2018). China has made remarkable improvement in air quality during recent years (Zhang et al., 2017; Zhao et al., 2017; China, 2018; Zheng et al., 2018), however, air pollution is still severe, making it the fourth-ranked healthy risk factor (Stanaway et al., 2018).

Surface $PM_{2.5}$ and $O_3$ concentrations are determined by atmospheric processes of emissions, transport and dispersion, chemical transformation (due to gas-phase, aqueous-phase and aerosol chemistry), and dry and wet deposition. These processes are affected by meteorological conditions. Studies have shown that the surface $O_3$ and $PM_{2.5}$ concentrations are sensitive to different meteorological parameters. For example, Dawson et al. (2007b) have investigated the sensitivity of surface $O_3$ to different meteorological parameters in the eastern United States (US) using the comprehensive air quality model with extensions ($CAM_X$). The results showed that



temperature (T) had the greatest influence on daily $O_3$-8h of 0.34 ppb $K^{-1}$, followed by
absolute humidity (AH) of 0.025 ppb $\%^{-1}$. Bernard et al. (2001) also confirmed that T
presented a notable positive correlation with the surface $O_3$ concentration. The effects
of meteorological parameters on $PM_{2.5}$ are even more complicated. Tran and Mölders
(2011) showed that elevated $PM_{2.5}$ concentrations tended to occur under the condition
of calm wind, low T and relative humidity in Fairbanks, Alaska. Olvera Alvarez et al.
(2018) used a land use regression model to analyze the effects of different
meteorological parameters on $PM_{2.5}$ in El Paso, Texas and obtained the same
conclusion in winter, but in spring, the high $PM_{2.5}$ level was associated with high wind
speed (WS) and low humidity. Dawson et al. (2007a) studied the effects of individual
meteorological parameters in the Eastern US and found that $PM_{2.5}$ concentration
decreased markedly as the increased precipitation (PCP) in winter, but in summer, the
main meteorological factors affecting the $PM_{2.5}$ concentration were T, WS and
planetary boundary layer height (PBLH). Dawson et al. (2009) simulated the effects
of climate change on regional and urban air quality in the Eastern US, and found
$PM_{2.5}$ concentration decreased by 0.3 μg $m^{-3}$ in January mostly due to increasing in
PCP and increased by 2.5 μg $m^{-3}$ in July largely due to decreasing in PBLH and WS.
Horne and Dabdub (2017) altered various meteorological parameters to investigate
their effects on $O_3$, $PM_{2.5}$ and secondary organic aerosols (SOA), and found that the T
predominated the effects of meteorology in California.
Many studies have proved that meteorological conditions play very important
roles in air pollution events in China. Studies found that the pollutant concentrations





could vary up to several times, due to meteorological changes with the same emission
sources (Zhang et al., 2010; Xing et al., 2011; Zheng et al., 2015; Cai et al., 2017; Liu
et al., 2017; Ning et al., 2018; Yang et al., 2018; Li et al., 2019b). For example, Xing
et al. (2011) studied that the difference between the effects of 2007 and 2008
meteorological conditions on air quality during the 2008 Beijing Olympics. They
found higher humidity in August 2008 was beneficial to the formation of $SO_4^{2-}$ by up
to ~60%, and lower T prevented the evaporation of $NO_3^-$ by up to ~60%. Liu et al.
(2017) reported that the monthly mean $PM_{2.5}$ concentrations in the Jing-Jin-Ji (JJJ)
area in December 2015 increased by 5%~137% due to the unfavorable weather
conditions such as low WS and high humidity.

102       A few studies investigated the relationships between air quality and

meteorological conditions in China. Zhang et al. (2015) conducted a correlation
analysis between air quality and meteorology in three megacities Beijing, Shanghai
and Guangzhou in China. The result showed that air pollutants were significantly
negatively correlated with WS, and $O_3$ had a positive correlation with T. Yin et al.
(2016) found that the relationship between WS and $PM_{2.5}$ has complicated influence,
with higher PM at low and high WS than in light to moderate winds in Beijing from
2008 to 2014. Xu et al. (2018) examined the variations of $PM_{2.5}$ concentration in
January 2017 in China compared to that in January 2016 and found meteorological
conditions of low WS, high humidity, low PBLH and low PCP contributed to $PM_{2.5}$
concentration worsening by 29.7%, 42.6% and 7.9% in the JJJ region, the Pearl River
Delta (PRD) region and the Cheng-Yu Basin (CYB) region, respectively. Ma et al.


(2019) analyzed the effects of meteorology on air pollution in the Yangtze River Delta
(YRD) region during 2014-2016 and found $PM_{2.5}$ was highly negatively correlated to
WS, while $O_3$ concentration was positively correlated to T but negatively related to
relative humidity. Zhu et al. (2017) reported that the surface concentrations of $O_3$
increased by 2-6 ppb in January and 8-12 ppb in July 2014 in PRD, mainly due to the
increase in T and the decrease in NOx emissions.
These studies have investigated the impacts of meteorological conditions on
$PM_{2.5}$ and $O_3$ in certain regions of China, however, quantitative sensitivity of $PM_{2.5}$
and $O_3$ to meteorological parameters has not been examined. The objective of this
study is to quantify the sensitivity of $O_3$ and $PM_{2.5}$ to different meteorological
parameters in winter and summer in different regions of China. The paper is
constructed as following, Section 2 describes the method used to estimate the
sensitivity, and Section 3 presents the effects of each meteorological variable on $O_3$
and $PM_{2.5}$ in China and in five representative cities. Conclusions are then summarized
in Section 4.

## 2. Methods

The sensitivity of $O_3$ and $PM_{2.5}$ associated with changes in meteorological
parameters was quantified using the Community Multiscale Air Quality (CMAQ)
model version 5.0.2. The meteorological parameters include T, WS, AH, PBLH, PCP,
and cloud liquid water content (CLW). A base case was firstly simulated with
meteorological fields predicted by the Weather Research and Forecasting (WRF)
model v3.7.1 (http://www.wrf-model.org/) using the NCEP FNL Operational Model



Global Tropospheric Analyses dataset as the initial and boundary conditions. The base
case has been described in a previous study and the model configurations of the base
case were reported there (Hu et al., 2015). The WRF predicted meteorological
parameters and the CMAQ predicted surface $O_3$ and $PM_{2.5}$ have been evaluated
against observations at 422 sites in 60 major cities in China, and the accuracy of the
model performance has been validated (Hu et al., 2016).
A suit of perturbation scenarios was created, and in each scenario, a certain
meteorological parameter was perturbed to a certain extent. The details of the
perturbation scenarios are listed in Table 1. Among those changes, the T was absolute
changes, and other parameters are relative variations. Then the CMAQ model was
re-run to predict the air quality under the perturbed meteorological condition. The
emissions and other inputs were kept unchanged in each perturbed meteorological
scenarios, therefore the difference of $O_3$ and $PM_{2.5}$ concentrations between each of the
perturbation case and the base case was due to the change in the specific
meteorological parameter, and the sensitivity of $O_3$ and $PM_{2.5}$ to individual
meteorological parameters could be quantitatively determined.
The modeling domain covers East Asia, including entire China, with a horizontal
resolution of $36 \times 36$ km$^2$. The base case and perturbation cases were conducted in
January and July in 2013, representing the winter and the summer conditions,
respectively. In addition to the regional analysis, five representative megacities were
selected, i.e., Beijing, Shanghai, Guangzhou, Chongqing, Xi'an (Fig.1). These cities
are located in the North China Plain (NCP), YRD, PRD, CYB, and Guanzhong Plain,



respectively, where serious air pollution problems often occur. In this study, $O_3$-8h
was used in the $O_3$ analyses, and 24h average $PM_{2.5}$ was used in the $PM_{2.5}$ analyses, if
not specifically stated.

## 3. Results and Discussion

### 3.1 Impacts of meteorological parameters on surface $O_3$

Figs. 2(a) and 2(b) show the spatial-distribution of the predicted monthly average
$O_3$-8h concentrations in January and July, respectively. In January, the highest average
concentrations are about 70 ppb in the Sichuan Basin, and the concentrations in
southern and eastern China are generally higher than those in northern China. In July,
the highest average concentrations are over 80ppb in the large areas of NCP and YRD,
CYB, and Guangzhou areas in the PRD.
Fig. 3 shows the spatial distribution of the concentration changes of $O_3$-8h in
January and July due to change of T + 1.0 K, WS - 10%, AH +10% PBLH -
20%,CLW + 10% case, and PCP + 10%, respectively. Fig. S1-S3 shows the results
due to other extent changes in these parameters. When T increases 1.0 K (Fig. 3(a)),
$O_3$-8h increases 1-2 ppb in most area of eastern and central China in January and in
NCP and YRD in July, which is consistent with the high $O_3$ spatial distribution in the
base case (shown in Fig. 2). $O_3$-8h decreases up to 4 ppb in January in Northeast
China, and up to 2 ppb in the Southwest border of China and the East China Sea,
which are the areas of low $O_3$ concentrations (generally less than 35 ppb). Therefore,
the effect of T on $O_3$ is dependent on the $O_3$ formation regime. An increase in T
promotes $O_3$ formation chemistry in net $O_3$ formation areas, but accelerates $O_3$





consumption chemistry in the net $O_3$ loss areas.
Fig. 3(b) shows that the differences of $O_3$-8h in January and July when WS is 10%
less than the base case in 2013. The influence of wind on $O_3$ concentration is complex,
but generally, slower WS decreases $O_3$ in January in most parts of China, particularly
in Sichuan by up to 3 ppb, but increases $O_3$ in July by a few ppb over the most areas
in eastern and central China. Therefore, the impact of WS on $O_3$ appears opposite in
winter and summer. Weaker winds slow down the dispersion of NOx and VOCs,
which is conducive to $O_3$ formation in summer when the vertical mixing is strong, but
increases $O_3$ titration in the surface in winter due to weaker vertical mixing.
Fig. 3(c) displays that the surface $O_3$ is expected to decrease generally less than 1
ppb when AH increases by 10% (relative change) both in January and July in most
land areas of China except in the northeast area. Fig. 3(d) shows that a 20% decrease
of PBLH leads to $O_3$-8h decreases by a few ppb in most area in January, while in July
$O_3$-8h increases in eastern and central regions, especially in YRD, CYB and areas in
Hubei-Hunan-Jiangxi in the central China. Sensitivity of $O_3$ to CLW and PCP is
relatively small. Fig. 3(e) demonstrates that $O_3$-8h changes -0.03 ppb to 0.03 ppb in
January and July for a 10% increase in CLW. Fig. 3(f) demonstrates that a 10%
increase in PCP results in -0.1 to 0.2 ppb changes in $O_3$-8h. $O_3$ changes due to the six
meteorological factors with different extents of perturbation (Figs. S1-S3) shows the
similar trends and spatial patterns.
**3.2 Impacts of meteorological parameters on surface $PM_{2.5}$**
Figs. 2(c) and 2(d) show the spatial distribution of the monthly average surface



$PM_{2.5}$ concentrations in January and July. $PM_{2.5}$ in January reaches over 200 µg m$^{-3}$ in
JJJ, SYB, central China, and urban areas in the Northeast China. $PM_{2.5}$ is much lower
in July, generally lower than 50 µg m$^{-3}$, but is high (up to 70 µg m$^{-3}$) in areas in the JJJ,
YRD, and central China regions.
Fig. 4 shows the spatial distribution of $PM_{2.5}$ changes due to the same changes of
meteorological factors, as in Fig. 3. The $PM_{2.5}$ results of other cases of the sensitivity
study are shown in Figs. S4-S6 of the Supplementary Materials. The results indicate
that in January, a 1.0 K increase in T leads to up to 5-6 µg m$^{-3}$ decrease of $PM_{2.5}$ in JJJ
and central China; in July, a 1.0 K increase in T causes $PM_{2.5}$ increase by about 1 µg
m$^{-3}$ in southern China but decrease by 1-3 µg m$^{-3}$ in the JJJ and east coast region. A 10%
decrease in WS causes $PM_{2.5}$ increase up to over 40 µg m$^{-3}$ in January and up to 5 µg
m$^{-3}$ in July. A 10% relative increase in AH leads to $PM_{2.5}$ increase of up to 6 µg m$^{-3}$ in
January and up to 2 µg m$^{-3}$ in JJJ and northeast regions but slightly decrease of less
than 1 µg m$^{-3}$ in southern China in July. A 20% decrease of PBLH causes $PM_{2.5}$
increases by up to 20 µg m$^{-3}$ in January and up to 4 µg m$^{-3}$ in July. The impact of
CLW and PCP on $PM_{2.5}$ is small, and generally increase in CLW increases surface
$PM_{2.5}$ and increase in PCP decreases $PM_{2.5}$.
The changes in the total $PM_{2.5}$ mass concentrations are determined by the
changes in the chemical components of $PM_{2.5}$. Fig. S7 displays the fraction of $PM_{2.5}$
species (elemental carbon (EC), primary organic carbon (POC), secondary organic
aerosol (SOA), sulfate ($SO_4^{2-}$), nitrate ($NO_3^-$), and ammonium ($NH_4^+$)) in five
representative cities. Secondary inorganic aerosols ($SO_4^{2-}$, $NO_3^-$, $NH_4^+$) are the major

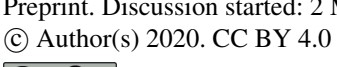



PM components, accounting for over 50% of $PM_{2.5}$ in January and about 40% in July.
Fig. 5 and Fig. 6 show the changes of the major $PM_{2.5}$ components due to the same
changes of meteorological factors as in Fig. 4 in January and in July, respectively. The
results show that the effects of the meteorological parameters on the total $PM_{2.5}$
(shown in Fig. 4) are mainly due to their effects on $SO_4^{2-}$, $NO_3^-$, and $NH_4^+$ in January,
and due to the changes in $SO_4^{2-}$, $NO_3^-$, $NH_4^+$, and SOA in July. In general, PBLH, WS,
and PCP are negatively correlated to $SO_4^{2-}$, $NO_3^-$, and $NH_4^+$ formation, but AH and
CLW are positively correlated to these components. SOA concentrations are much
higher in July than in January due to the contribution from biogenic emissions (Hu et
al., 2017a). SOA formation is affect by reaction rates (positively affected by T),
availability of oxidants (such as changes in $O_3$), and hydrogen ion strength (affected
by changes in $SO_4^{2-}$, $NO_3^-$, $NH_4^+$). SOA concentrations mainly increase in south
China.
It is worthwhile noting that the effects of T on $SO_4^{2-}$ and $NO_3^-$ (changes of $NH_4^+$
is determined by changes of $SO_4^{2-}$ and $NO_3^-$). Both in January and July, increase in T
decreases $SO_4^{2-}$ and $NO_3^-$ in the major areas of eastern China. The $NO_3^-$ decreases is
expected because volatile $NH_4NO_3$ favors more in gas phase in higher temperature,
and this result is consistent with studies in other regions (Dawson et al., 2007a;Horne
and Dabdub, 2017). $SO_4^{2-}$ is found to increase with T increase in those studies because
faster gas- and aqueous- phase reactions of $SO_4^{2-}$. However, our finding of $SO_4^{2-}$ in
China is opposite. The CMAQ-Sulfur Tracking Model (CMAQ-STM) was further
used to track the $SO_4^{2-}$ formation from different processes. The results confirm that





the $SO_4^{2-}$ production from gas- and aqueous- phase increases with T increase. But
meanwhile $SO_4^{2-}$ production from heterogeneous reactions is reduced more when T is
increased. Heterogeneous $SO_4^{2-}$ formation has been proposed as a major $SO_4^{2-}$
formation pathway during China haze events (Wang et al., 2014b; Gen et al., 2019;
Huang et al., 2019; Li et al., 2019a) and in this study it accounts for up to ~75% of
total $SO_4^{2-}$ production. The treatment of heterogeneous $SO_4^{2-}$ formation currently is
modeled as a surface-controlled uptake process, in which the formation rate is
determined by the aerosol surface area and the uptake coefficient of $SO_2$ on particle
surface (Ying et al., 2014). When T is increased, the particle surface area decreases (as
particle mass concentration decreases due to a combined effect of other components),
resulting in decrease in the heterogeneous $SO_4^{2-}$ formation.

## 3.3. Quantitative sensitivity of $O_3$ and $PM_{2.5}$ to individual

**meteorology parameters**
The quantitative sensitivity of $O_3$ and $PM_{2.5}$ concentrations to individual
meteorological parameter is calculated by linear fitting of the changes in
monthly-average concentrations under all of the six perturbed cases of the
meteorological parameter. Figs. S8-S10 in Supplementary Materials show the
calculation examples of T, WS, and AH on $O_3$ at the five major cities of Beijing,
Chongqing, Guangzhou, Shanghai and Xi'an, and Figs. S11-S13 shows the examples
for the $PM_{2.5}$ cases. Fig.7 demonstrates the sensitivities of $O_3$-8h and $PM_{2.5}$ and its
components to each meteorological parameter in the five cities. In January, T has a





positive impact on $O_3$ in all cities, and the largest impact is in Chongqing with a rate
of +1.69 ppb $K^{-1}$. In July, $O_3$ also shows a strong positive sensitivity to T in Beijing
with +1.06 ppb $K^{-1}$ and in Shanghai with +0.98 ppb $K^{-1}$, but has a small negative
sensitivity (-0.15 ppb $K^{-1}$) in Xi'an and a moderate negative sensitivity (-0.74 ppb $K^{-1}$)
in Guangzhou. The $O_3$ sensitivity to T in Guangzhou in July shows a highly nonlinear
trend and is very different from other cities (Fig. S8(c)). More studies are needed to
investigate the effects of T on $O_3$ pollution in the YRD region during summertime.
WS and PBLH both have positive effects on $O_3$-8h in January, the effects vary
significantly among cities, with 0.004-0.3 ppb $\%^{-1}$ for WS and 0.04-0.14 ppb $\%^{-1}$ for
PBLH. AH has a negative effect on $O_3$-8h in January, ranging from -0.01 to -0.15
ppb $\%^{-1}$. But in July, the impacts of WS, AH, and PBLH are negative in most cities,
with a range of -0.05 to -0.18, -0.05 to -0.13, and -0.02 to -0.07 ppb $\%^{-1}$, respectively.
Generally speaking, the sensitivity of $O_3$ to T is obviously higher than that of WS, AH,
and PBLH. The sensitivity of $O_3$ to CLW and PCP is even minimal (less than 0.01
ppb $\%^{-1}$) and mostly negative.

285        Negative sensitivities are found for surface $PM_{2.5}$ concentrations to T, WS, PBLH,

and PCP, and positive sensitivities for $PM_{2.5}$ to AH and CLW. The sensitivity of T in
the five cities ranges from -1.5 to -3.6 μg $m^{-3}$ $K^{-1}$ in January and -0.3 to -1.65 μg $m^{-3}$
$K^{-1}$ in July. $PM_{2.5}$ is also very sensitive to WS in January, with a range of -0.8 to -2.97
μg $m^{-3}$ $\%^{-1}$, while the sensitivity (-0.03 to -0.19 μg $m^{-3}$ $\%^{-1}$) becomes much smaller in
July. The sensitivity to PBLH is -0.12 to -0.58 μg $m^{-3}$ $\%^{-1}$ in January and -0.003 to
-0.23 μg $m^{-3}$ $\%^{-1}$ in July. The sensitivity to AH is 0.16 to 0.30 μg $m^{-3}$ $\%^{-1}$ in January



and 0.05 to 0.27 μg m$^{-3}$ %$^{-1}$ in July. Sensitivity to CLW and PCP is small in January
and July, mostly less than 0.01μg m$^{-3}$ %$^{-1}$. The PM$_{2.5}$ sensitivities can be explained by
the major components of SO$_4^{2-}$, NO$_3^-$, and NH$_4^+$ in January and by SO$_4^{2-}$, NO$_3^-$, NH$_4^+$,
and SOA in July.

296        Fig. 8 shows the spatial variations of the sensitivity of O$_3$-8h and PM$_{2.5}$ to the

meteorological parameters. The sensitivity of O$_3$-8h to temperature is more significant
in Sichuan and southern provinces of China in January, and in NCP and YRD in July,
up to +2 ppb K$^{-1}$ in both January and July. O$_3$-8h sensitivity to WS is diverse in space,
and is generally positive in Sichuan and southern provinces in January; and it is
negative in east China but positive in west China. O$_3$-8h sensitivity to AH is generally
negative in both months in most regions of China, except the northeast in January and
southwest in July. O$_3$-8h sensitivity to PBLH is mostly positive in January but
becomes negative in YRD, CYB, NCP, and central China in July. O$_3$-8h sensitivity to
CLW and PCP is negligible.

306        Fig. 9 displays the spatial variations of the sensitivity of surface PM$_{2.5}$ to the

meteorological parameters. PM$_{2.5}$ sensitivities to the meteorological parameters are
more consistent in January and July than the cases of O$_3$, i.e., negative sensitivity to T,
WS, PBLH, PCP, and positive to AH and CLW in most regions of China in both
months. On the other hand, PM$_{2.5}$ sensitivities are more profound in January than in
July. PM$_{2.5}$ sensitivity to T is up to -5 μg m$^{-3}$ K$^{-1}$ in January and up to -2 μg m$^{-3}$ K$^{-1}$ in
July. PM$_{2.5}$ sensitivity to WS is up to -3 μg m$^{-3}$ %$^{-1}$ in January, and up to -0.4 μg





$m^{-3}$ %$^{-1}$ in July. $PM_{2.5}$ sensitivity to PBLH is up to -1 $\mu g$ $m^{-3}$ %$^{-1}$ in January, and up to
-0.14 $\mu g$ $m^{-3}$ %$^{-1}$ in July. $PM_{2.5}$ sensitivity to AH is up to +0.6 $\mu g$ $m^{-3}$ %$^{-1}$ in January,
and up to 0.3 $\mu g$ $m^{-3}$ %$^{-1}$ in July. The sensitivities to CLW and PCP is small, compared
to the other four meteorological parameters. $PM_{2.5}$ sensitivity to T is negative in most
land areas of China in January and in NCP and YRD in July because of the negative
effects of T on $SO_4^{2-}$, $NO_3^-$, and $NH_4^+$, as discussed in previous sections. $PM_{2.5}$
sensitivity to T is positive in south China in July due to more SOA with higher T.

## 4. Conclusions

Meteorological conditions can have a great influence on surface $O_3$ and $PM_{2.5}$
concentrations. In this study, the sensitivities of $O_3$-8h and $PM_{2.5}$ to T, WS, AH, PBLH,
PCP, and CLW are quantitatively estimated in January and July, respectively in China.
$O_3$-8h is most sensitive to T and the sensitivity can be up to +2 ppb K$^{-1}$ in both
January and July, and the sensitivity is dependent on the $O_3$ chemistry formation or
loss regime, i.e., positive in the net $O_3$ formation areas, and negative in the $O_3$
consumption areas. In general, $PM_{2.5}$ sensitivities are negative to T, WS, PBLH, and
PCP and positive to AH and CLW in most regions of China in both January and July.
The sensitivities in January are much larger than in July. $PM_{2.5}$ sensitivities to T, WS,
AH, and PBLH are important. The $PM_{2.5}$ sensitivities to these meteorological
parameters are through major effects on $SO_4^{2-}$, $NO_3^-$, $NH_4^+$, and SOA. The
sensitivities of $O_3$ and $PM_{2.5}$ to CLW and PCP are negligible. The results show that $O_3$
and $PM_{2.5}$ concentrations in China are greatly affected by meteorological conditions,
therefore changes in these meteorological parameters due to climate change or



inter-annual meteorological variations could potentially alter $O_3$ and $PM_{2.5}$
concentrations significantly, and it should consider these effects when developing
emission control strategies. The results also show that the $O_3$ and $PM_{2.5}$ sensitivities to
meteorological parameters have substantial spatial variations. Future studies can
further investigate how the changes in meteorological conditions affect the
effectiveness of emission control plans in reaching the designed air quality objectives
in the different regions of China.

**Data availability**. All of the modeling results will be available online after we
publish the paper.
**Author contributions.** ZS, and JH designed research. ZS, LH, JL, QY, HZ
and JH contributed to model development and configuration. ZS, LH, JL, and JH
analyzed the data. ZS prepared the manuscript and all coauthors helped improve the
manuscript.
**Competing interests.** The authors declare that they have no conflict of
interest.
**Acknowledgment**
This work was supported by the National Key R&D Program of China
(2018YFC0213800), the National Natural Science Foundation of China (41975162,
41675125 and 41705102), and Jiangsu Environmental Protection Research Project

355 (2016015).




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





Tables and Figures

Table 1. Meteorological perturbations imposed in this study

| Meteorological Parameter | Changes in Values Examined |
|---|---|
| Temperature (T) | ±0.5K,±1.0K,±1.5K |
| Wind speed (WS) | ±5%,±10%,±20% |
| Absolute Humidity (AH) | ±5%,±10%,±20% |
| Boundary layer height (PBLH) | ±10%,±20%,±30% |
| Cloud liquid content (CLW) | ±5%,±10%,±20% |
| Precipitation (PCP) | ±5%,±10%,±20% |







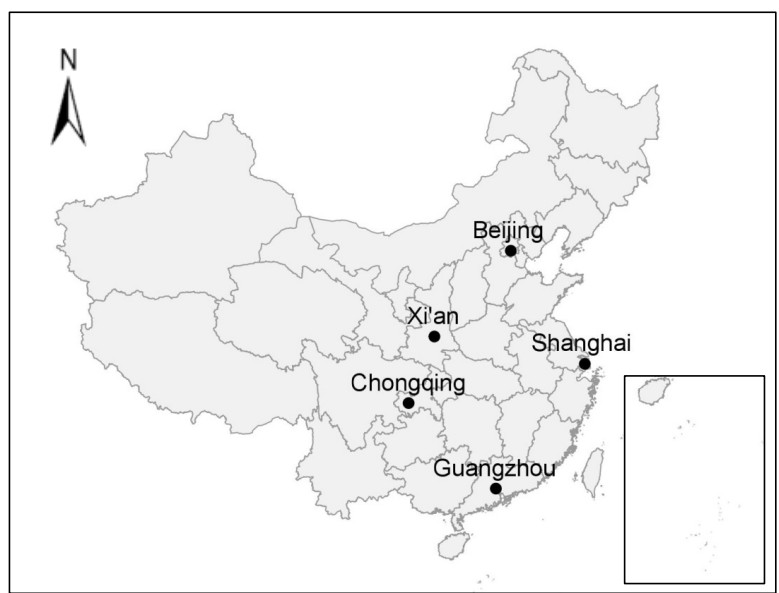


**Fig.1** Location map of China and the five cities.

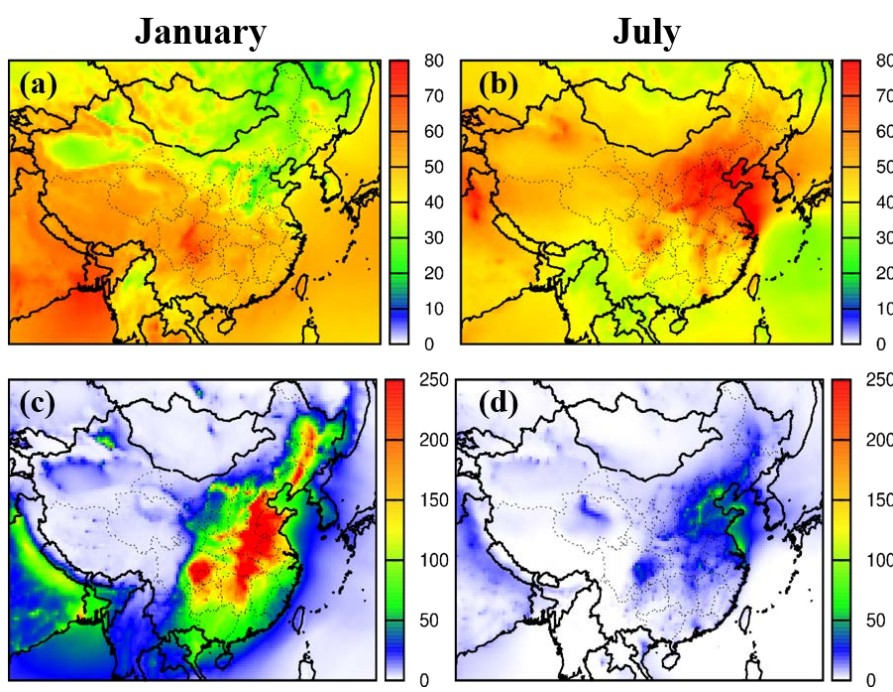


**Fig.2** Spatial distributions of monthly average $O_3$-8 h (ppb) in (a) January and (b) July, and
monthly average $PM_{2.5}$ ($\mu g\ m^{-3}$) in (c) January and (d) July 2013.


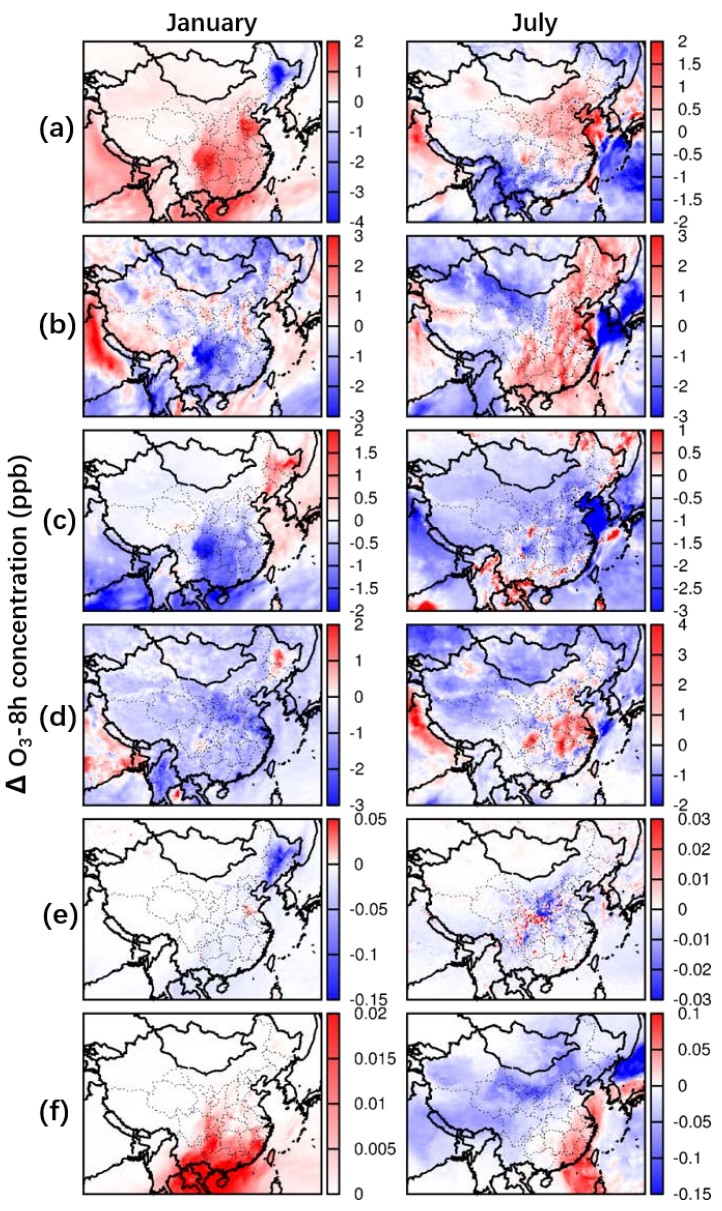

**Fig.3** Changes in monthly average O$_3$-8h (ppb) in January and July, 2013 due to (a) T+1.0K, (b) WS-10%, (c) AH+10%, (d) PBLH-20%, (e) CLW+10%, and (f) PCP+10%.

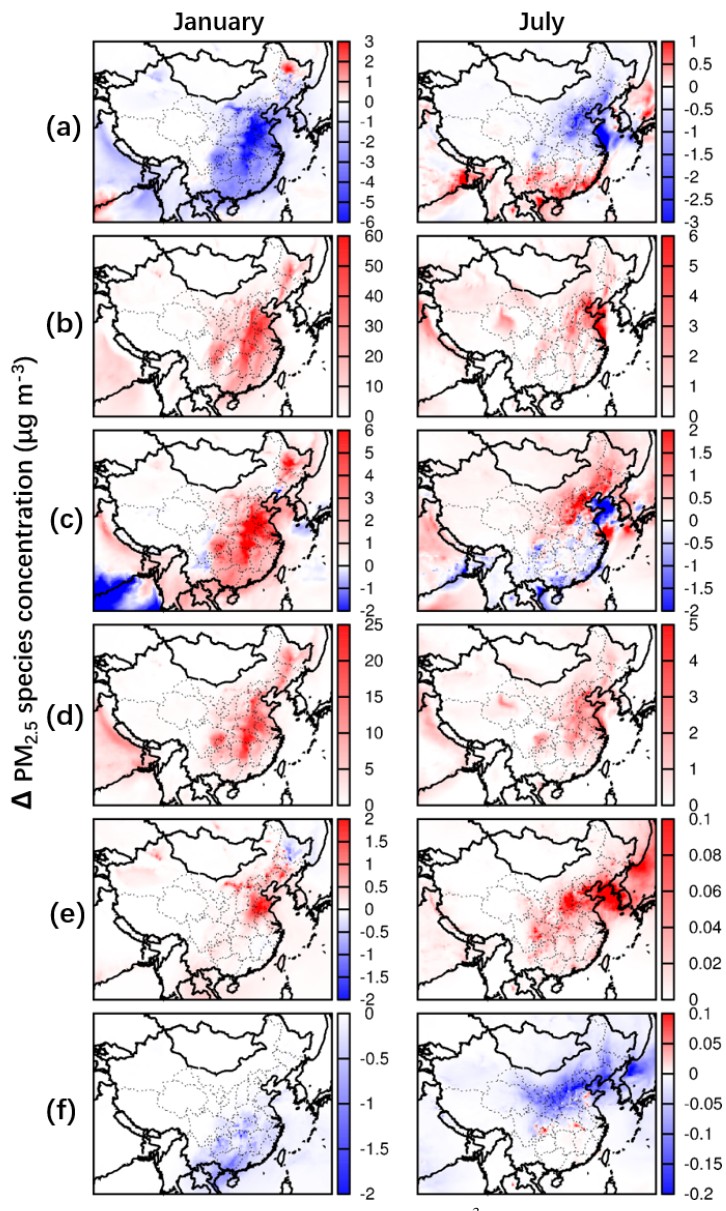

**Fig.4** Changes in monthly average PM$_{2.5}$ concentration (μg m$^{-3}$) in January and July, 2013 due to (a) T+1.0K, (b) WS-10%, (c) AH+10%, (d) PBLH-20%, (e) CLW+10%, and (f) PCP+10%.

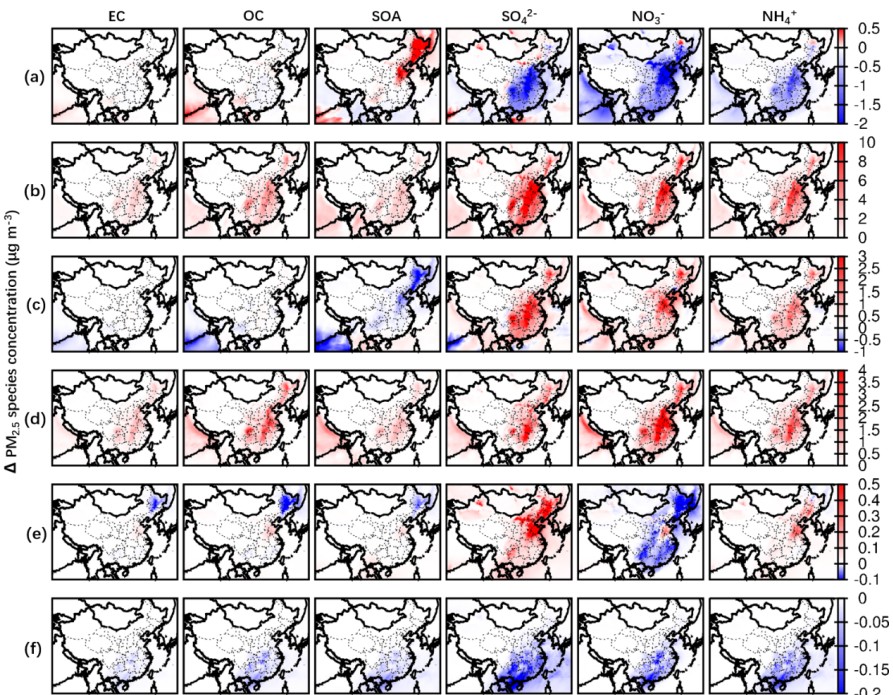


**Fig.5** Changes in monthly average PM$_{2.5}$ component concentration (μg m$^{-3}$) in January due to (a)
T+1.0K, (b) WS-10%, (c) AH+10%, (d) PBLH-20%, (e) CLW+10%, and (f) PCP+10%.





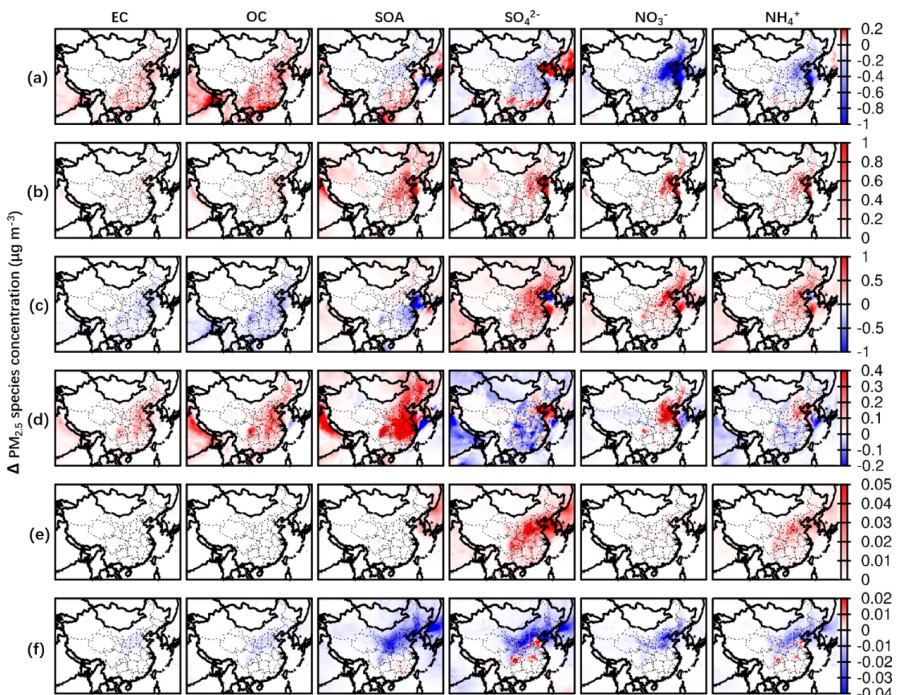


**Fig.6** Changes in monthly average PM$_{2.5}$ component concentration (μg m$^{-3}$) in July due to (a)
T+1.0K, (b) WS-10%, (c) AH+10%, (d) PBLH-20%, (e) CLW+10%, and (f) PCP+10%.




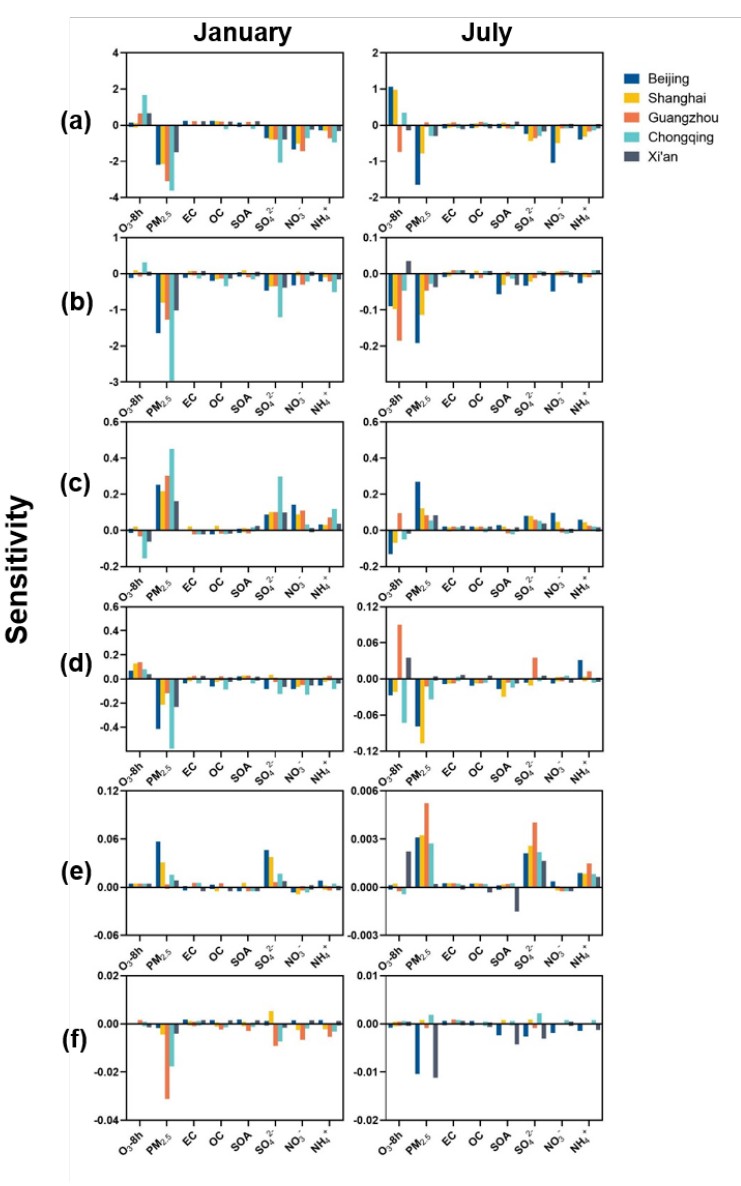

**Fig.7** Sensitivity of $O_3$-8h, $PM_{2.5}$ and its components to meteorological parameter of (a) T, (b) WS, (c) AH, (d) PBLH, (e) CLW, and (f) PCP in five cities in China. The unit of sensitivity is ppb $K^{-1}$ for $O_3$-8h to T, and is ppb $\%^{-1}$ for $O_3$-8h to other meteorological parameters; and the unit is $\mu g\ m^{-3}$ $K^{-1}$ for $PM_{2.5}$ and its components to T, and is $\mu g\ m^{-3}\ \%^{-1}$ for $PM_{2.5}$ and its components to other meteorological parameters.






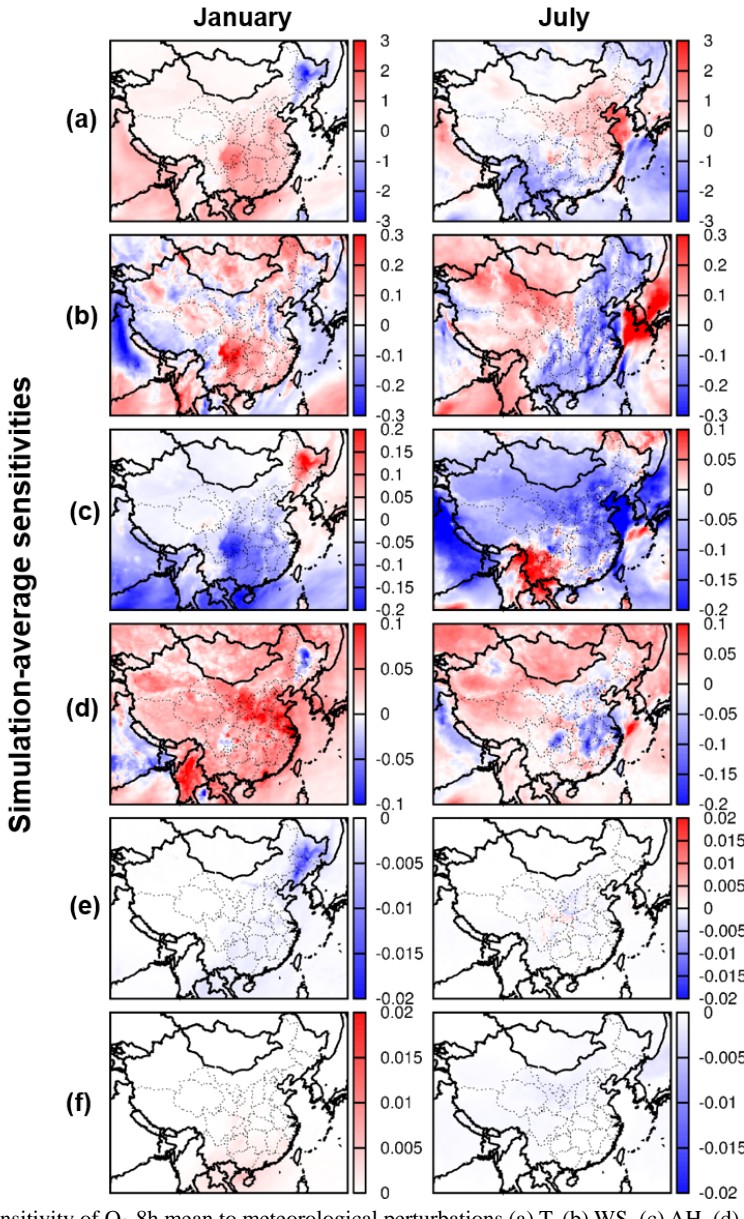


**Fig.8** Sensitivity of $O_3$-8h mean to meteorological perturbations (a) T, (b) WS, (c) AH, (d) PBLH, (e) CLW, (f) PCP in China. The value in T is measured in ppb $K^{-1}$, and others is ppb $\%^{-1}$.



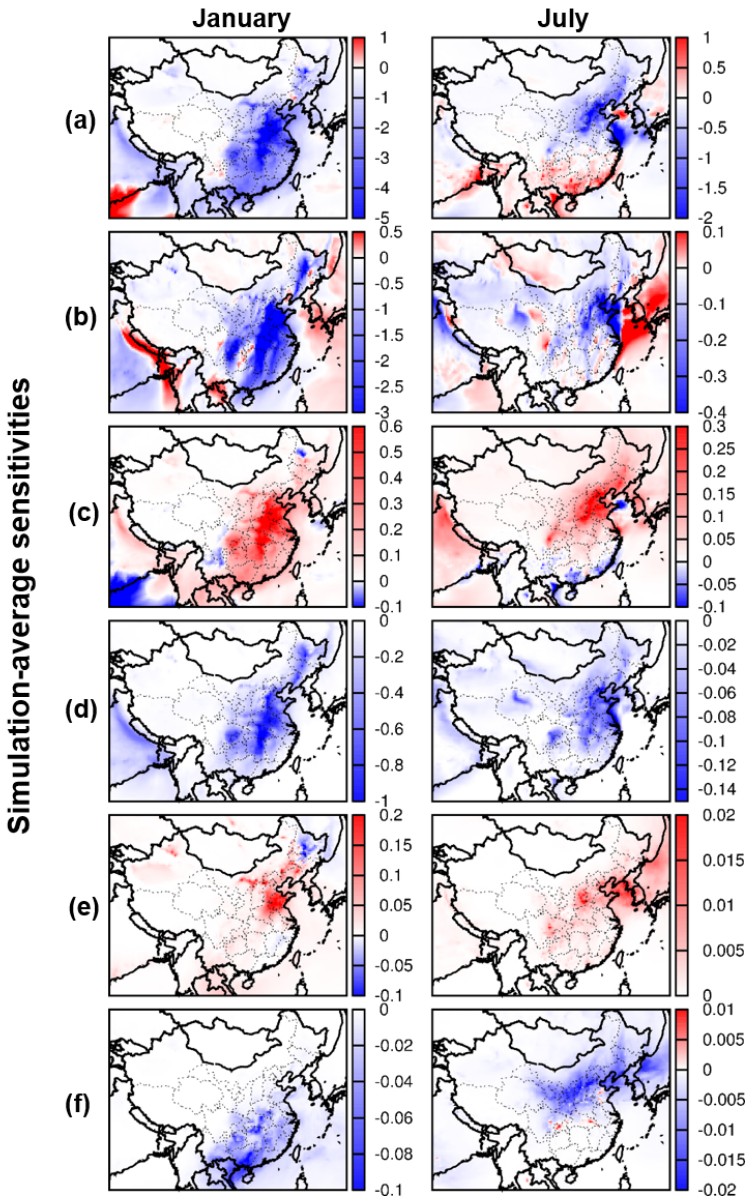

**Fig.9** Sensitivity of PM$_{2.5}$ mean to meteorological perturbations (a) T, (b) WS, (c) AH, (d) PBLH,
(e) CLW, (f) PCP in China. The value in T is measured in μg m$^{-3}$ K$^{-1}$, and others is μg m$^{-3}$ %$^{-1}$.