# Peer review of "Sensitivity Analysis of the Surface Ozone and Fine"

_Atmospheric Chemistry and Physics, 2020_

## Referee Comment (RC1) · Anonymous Referee #1 · 20 Apr 2020

The paper discusses the sensitivity of surface ozone and PM2.5 in China to meteorological parameters. The information presented in the paper is useful to understand the interaction between pollution and meteorology, and regional difference in the sensitivity of emission control measures. I'd recommend the publication of the paper if the following comments are addressed:

(1) The method description is very brief, and the details in implementation may affect the interpretation of the results. In particular, I see one difficulty in this type of sensitivity simulation that a simple perturbation of individual parameters may lead to unphysical meteorological fields. For example, increasing/decreasing T by 1 K under some conditions may turn saturated/unsaturated air into unsaturated/saturated, but since only T is perturbed, no cloud is dissipated/formed in response to changing T. Another example, a simple perturbation of wind speed may generate a wind field that violates the physics, and is inconsistent with the pressure field that feeds into the air quality simulation, which may lead to spurious sensitivities in the result. Even more difficult is to perturb wind direction, though I notice the authors did not assess the wind direction sensitivity. In general, I'd like to see if and how this type of issues is handled by the authors. The current method description is too brief to tell the exact implementation. Other useful details to include are if the perturbations are done for the entire atmosphere or only in the boundary layer, if they are done for the whole day uniformly or only in the daytime.

(2) The responses of emissions to meteorological parameters are not included in the assessment. The responses of emissions to meteorology is a significant contributor to the overall meteorological sensitivity of ozone and PM2.5. To name a few, the effect of T on biogenic emissions, the effect of T on soil NOx emissions, the cloud cover/convection on lightning NOx emissions, the effect of T on power plant NOx emissions (high T leads to higher electricity demand in summer). Because emissions are held unchanged in the simulations, these effects are not included, which makes the analysis incomplete and less informative. This caveat needs to be discussed in the paper.

(3) Evaluation against observations. The O3-T slope from model simulations is often found to be much lower than that derived from observations, suggesting that model tends to underestimate the sensitivity of O3 to meteorology. The current paper provides no evaluations of how good the model in use could reproduce the observed chemical-met relationship. Note this evaluation is different from evaluation of chemical concentrations, and is perhaps more relevant for the current work.

(4) In abstract and elsewhere (such as Line 282), the authors compare the different sensitivities. For instance, the paper says in Line 282 that "the sensitivity of O3 to
T is obviously higher than that of WS, AH, and PBLH". This is to compare apples to oranges, because these sensitivities are in different units!

The delta concentrations of O3 or PM2.5 from two simulations apparently depend on how much you perturb, and it is meaningless to compare which one is bigger unless the perturbations are carefully defined to relate to the variations of individual parameters.

---

## Referee Comment (RC2) · Anonymous Referee #2 · 26 Apr 2020

The authors here have studied the effect of perturbation of meteorology on PM2.5 and O3 concentration across China. Overall, the manuscript was well written, method is sound, results are valid. I would recommend it to be published after addressing the following issues.

1. The authors here conduct the sensitivity analysis by perturbing the value of one meteorological parameter and keeping value of others constant. In real world when a meteorological parameter changes, a corresponding change in other parameters also takes place. This will affect the entire results.

2. Even if we assume that the authors are trying to only depict the sensitivity of PM2.5

and O3 on perturbation of meteorological parameters, the above said knowledge won't be handy to the authorities when trying to implement emission control in such scenarios. Since perturbation of one meteorological parameter will result in corresponding change in other parameters and since the current simulation is only based on assumption that only one parameter will change at any given time, the results from current sensitivity analysis won't be of any use.

3. Solar radiation apart from temperature is also one of the main factors affecting O3 why haven't the authors studied sensitivity of O3 concentration to change in solar radiation.

4. The authors doesn't mention on what basis they change the meteorological parameters i.e. on what basis is the magnitude of change in parameters considered.

5. Line 179-182, the authors discuss regarding effect of Temperature on Ozone in Ozone forming regime. Any references to suggest that the said areas in China are in ozone forming or ozone consumption regimes?

6. In Figures S8-S13, the authors estimate the quantitative sensitivity of O3 and PM2.5 concentrations to change in individual meteorological parameters by linear fitting of the changes. The authors should also report the corresponding R-squared, slope and significance values, it would help to understand the rate of change of PM2.5 or O3 per change in meteorological parameters and if at all the rate of change is statistically significant.

7. Does the authors perturb meteorology parameters only for China in the domain? As per spatial variation figures, the domain also constitutes parts of south-east Asia?

**ACPD**

---

## Author Comment (AC1) · 12 Jun 2020

Referee #2 The authors here have studied the effect of perturbation of meteorology on PM2.5 and O3 concentration across China. Overall, the manuscript was well written, method is sound, results are valid. I would recommend it to be published after addressing the following issues.

1. The authors here conduct the sensitivity analysis by perturbing the value of one meteorological parameter and keeping value of others constant. In real world when a meteorological parameter changes, a corresponding change in other parameters also takes place. This will affect the entire results.

[Figure]

Responses: For 'real world' meteorology changes, climate/weather forecasting models are usually utilized to predict how an entire set of meteorological parameters will change under certain scenarios and to estimate the impacts on air quality. Despite the large uncertainties in predicting 'real world' climate changes, another problem with this method is that it is impossible to isolate the effects of individual meteorological parameters. Sensitivity studies are commonly used to achieve this objective by perturbing one parameter at a time and keeping other parameters unchanged. This method may not reflect 'real world' changes, but can provide information that the first method cannot provide, and this method has been applied in several studies, such as Dawson et al. (2007a), Dawson (2007b), and Horne et al. (2017).

2. Even if we assume that the authors are trying to only depict the sensitivity of PM2.5 and O3 on perturbation of meteorological parameters, the above said knowledge won't be handy to the authorities when trying to implement emission control in such scenarios. Since perturbation of one meteorological parameter will result in corresponding change in other parameters and since the current simulation is only based on assumption that only one parameter will change at any given time, the results from current sensitivity analysis won't be of any use.

Responses: The results are useful for implementing emission controls in several aspects. First, the results help identify the major meteorological factors to which PM2.5 and O3 have the largest sensitivities. For example, our results indicate that in July O3 is very sensitive to temperature but not so sensitive to PBL height in Beijing. Therefore, additional emission controls would be needed if temperature is predicted to increase in future, but not necessary if PBL height is predicted to increase (while temperature is predicted no significant increase). Second, the results show that the PM2.5 sensitivities to these meteorological parameters are mainly through secondary components (SO42-, NO3-, NH4+, and SOA). Therefore, more emission controls on the precursors of the secondary components would be needed in future to overcome the adverse impacts of meteorological condition changes on PM2.5. Third, this study aims to isolate

the effects of individual meteorological parameters on air quality. It is very straightforward to quantify the combined effects of changes in several meteorological parameters. As an example, we conducted an additional simulation to test the impact of all perturbations (T+1.0K, WS-10%, AH+10%, PBLH-20%, CLW+10%, and PCP+10%) on O3 and PM2.5 in January and July, and the results was shown in Fig.S8 in the revised manuscript.

3. Solar radiation apart from temperature is also one of the main factors affecting O3 why haven't the authors studied sensitivity of O3 concentration to change in solar radiation.

Responses: Solar radiation affects photolysis rates. In CMAQ, the photolysis rates are calculated in-line. First the clear-sky photolysis rates are calculated using the clear-sky actinic flux. Then photolysis rates are corrected to account for the effects of cloud and particle extinction. The actinic flux is calculated in real time as a function of time of day, longitude, latitude, altitude, and season, therefore is not perturbed in this study.

4. The authors doesn't mention on what basis they change the meteorological parameters i.e. on what basis is the magnitude of change in parameters considered.

Responses: The magnitude ranges of perturbations are based on IPCC AR5 report and the study of Dawson et al. (2007) and the references therein. For each parameter, three positive and three negative perturbations were then designed within its range to have a more comprehensive examination on the sensitivity of PM2.5 and O3 to this parameter. We add the above information in the method section.

5. Line 179-182, the authors discuss regarding effect of Temperature on Ozone in Ozone forming regime. Any references to suggest that the said areas in China are in ozone forming or ozone consumption regimes?

Responses: The net O3 formation areas and the net O3 loss areas are classified based on the O3 concentrations (shown Fig. 2). The background O3 is about 35 ppb,

therefore, areas with O3 concentrations over 35 ppb is the net O3 formation areas, and areas with O3 concentrations less than 35 ppb is the net O3 loss areas. We added the explanation in the revised manuscript.

6. In Figures S8-S13, the authors estimate the quantitative sensitivity of O3 and PM2.5 concentrations to change in individual meteorological parameters by linear fitting of the changes. The authors should also report the corresponding R-squared, slope and significance values, it would help to understand the rate of change of PM2.5 or O3 per change in meteorological parameters and if at all the rate of change is statistically significant.

Responses: Thanks for your suggestion. We added these metrics in Fig. S8-S13.

7. Does the authors perturb meteorology parameters only for China in the domain? As per spatial variation figures, the domain also constitutes parts of south-east Asia?

Responses: All perturbations were implemented uniformly in space on the modeling domain and in time through the modeling periods. The perturbations on temperature, wind speed, and absolute humidity were made in all layers. We have added above explanation in the method section.

References

Dawson, J., Adams, P., Pandis, S., 2007a. Sensitivity of PM 2.5 to climate in the Eastern US: a modeling case study. Atmospheric chemistry and physics 7, 4295-4309. Dawson, J.P., Adams, P.J., Pandis, S.N., 2007b. Sensitivity of ozone to summertime climate in the eastern USA: A modeling case study. Atmospheric environment 41, 1494-1511. Horne, J.R., Dabdub, D., 2017. Impact of global climate change on ozone, particulate matter, and secondary organic aerosol concentrations in California: A model perturbation analysis. Atmospheric Environment 153, 1-17 Rasmussen, D., et al., 2012. Surface ozone-temperature relationships in the eastern US: A monthly climatology for evaluating chemistry-climate models. Atmospheric Environment. 47,

142-153.

---

## Author Comment (AC2) · 12 Jun 2020

The paper discusses the sensitivity of surface ozone and PM2.5 in China to meteorological parameters. The information presented in the paper is useful to understand the interaction between pollution and meteorology, and regional difference in the sensitivity of emission control measures. I'd recommend the publication of the paper if the following comments are addressed: (1) The method description is very brief, and the details in implementation may affect the interpretation of the results. In particular, I see one difficulty in this type of sensitivity simulation that a simple perturbation of individual parameters may lead to unphysical meteorological fields. For example, increas-

ing/decreasing T by 1 K under some conditions may turn saturated/unsaturated air into unsaturated/saturated, but since only T is perturbed, no cloud is dissipated/formed in response to changing T. Another example, a simple perturbation of wind speed may generate a wind field that violates the physics, and is inconsistent with the pressure field that feeds into the air quality simulation, which may lead to spurious sensitivities in the result. Even more difficult is to perturb wind direction, though I notice the authors did not assess the wind direction sensitivity. In general, I'd like to see if and how this type of issues is handled by the authors. The current method description is too brief to tell the exact implementation. Other useful details to include are if the perturbations are done for the entire atmosphere or only in the boundary layer, if they are done for the whole day uniformly or only in the daytime.

Responses: To clarify how we perturb the meteorological parameters, we added the following sentences in the method section:

"All perturbations were implemented uniformly in space on the modeling domain and in time through the modeling periods. The perturbations on temperature, wind speed, and absolute humidity were made in all layers. To separate the effects of individual meteorological parameters, only one parameter was changed in each case while all other parameters were kept unchanged. Therefore, cloud dissipating or forming in response to changing temperature was not considered in the simulations. When perturbing horizontal wind speed, to avoid unphysical situations that mass would not be conserved, the vertical wind speed was adjusted in the vertical transport calculation based on the air density changes to conserve mass."

(2) The responses of emissions to meteorological parameters are not included in the assessment. The responses of emissions to meteorology is a significant contributor to the overall meteorological sensitivity of ozone and PM2.5. To name a few, the effect of T on biogenic emissions, the effect of T on soil NOx emissions, the cloud cover/convection on lightning NOx emissions, the effect of T on power plant NOx emissions (high T leads to higher electricity demand in summer). Because emissions are

held unchanged in the simulations, these effects are not included, which makes the analysis incomplete and less informative. This caveat needs to be discussed in the paper.

Responses: Thanks for the comments. In the method section, we added the following sentences:

"It is worthwhile to note that some meteorological parameters could have significant impacts on emissions, such as the effect of T on biogenic VOC and soil NOx emissions, the cloud cover/convection on lightning NOx emissions, the effect of T on power plant NOx emissions (high T leads to higher electricity demand in summer), which would affect air quality. Therefore, the sensitivities in this study only include the 'direct' effects of individual meteorological parameters on air quality. A full evaluation of the impacts of climate/weather changes on air quality should consider effects of the emissions changes."

(3) Evaluation against observations. The O3-T slope from model simulations is often found to be much lower than that derived from observations, suggesting that model tends to underestimate the sensitivity of O3 to meteorology. The current paper provides no evaluations of how good the model in use could reproduce the observed chemical-met relationship. Note this evaluation is different from evaluation of chemical concentrations, and is perhaps more relevant for the current work.

Responses: Thank you for your valuable advice. We conducted the evaluation of the O3-T relationship, following the method in Rasmussen et al. (2012). We have no O3 observations in January (O3 observations became available from March 2013 in China), so we only evaluated the results in July in the five cities as in the manuscript. We found that CMAQ overestimated the O3-T relationship (CMAQ: 2.4 ppb/K vs. observation: 0.8 ppb/K, shown in Figure S1). Please note that we only have 1 month data and we use daily MDA8 O3 and daily maximum temperature in the evaluation, while a much more meaningful evaluation should be performed to use monthly averaged MDA8 O3 and monthly average temperature over a long-term period. We added above evaluation and discussion in the revised manuscript.

(4) In abstract and elsewhere (such as Line 282), the authors compare the different sensitivities. For instance, the paper says in Line 282 that "the sensitivity of O3 to T is obviously higher than that of WS, AH, and PBLH". This is to compare apples to oranges, because these sensitivities are in different units! The delta concentrations of O3 or PM2.5 from two simulations apparently depend on how much you perturb, and it is meaningless to compare which one is bigger unless the perturbations are carefully defined to relate to the variations of individual parameters.

Responses: We modified the descriptions about the comparison among different meteorological parameters because of the different unit problem.

References

Rasmussen, D., et al., 2012. Surface ozone-temperature relationships in the eastern US: A monthly climatology for evaluating chemistry-climate models. Atmospheric Environment. 47, 142-153.

---

## Author Response (AR2)

Referee #1

I have three questions about the manuscript.

1. The comparison of simulated and observed O3-T relationship for five cities is not done properly (Line 187; Fig S1). The authors derived the O3-T slope from daily data in all five cities in the scatter plot (Fig. S1). Thus derived slope may be determined mainly by the "systematic" differences among the five cities. Indeed, Fig.3 and Fig. 7 shows the local O3-T slope over most regions of China is smaller than 1 ppb/K, inconsistent with the CMAQ slope shown in Fig. S1 and Line 187 (2.4 ppb/K).

Responses: Thanks for your comment. We compared the simulated and observed O3-T relationship in the five cities individually. Fig. S1 and the corresponding discussions have been updated in the revised manuscript. The results show that CMAQ predicts positive $O_3$-T relationship in most cities except in Beijing, and the model tends to underestimated the daily $O_3$-T relationship except in Shanghai. The underestimation of $O_3$-T by the CMAQ model in this study is consistent with the findings in Rasmussen et al. (2012).

The results in Fig. S1 is different from those in Fig. 3 and Fig. 7, because they are in different time scales. The results in Fig. S1 are of daily MDA8 $O_3$ concentrations, while the results in Fig. 3 and Fig. 7 are of monthly average concentrations.

[Figure]

Fig. S1. Observed and predicted relationships between surface MDA8 $O_3$ (ppb) and daily $T_{max}$ (K) in the 5 cities in July 2013.

2. I find Section 3.3 and Fig. 8 and 9 redundant. The spatial distributions are essentially the same between Fig. 3 and Fig. 8, and between Fig. 4 and Fig. 9. I do not see the point of these figures and corresponding discussions.

Responses: I think the reviewer may misunderstood Section 3.3 and Fig. 8 and Fig. 9. Fig. 3 and Fig. 4 illustrate the concentration changes of $O_3$ and $PM_{2.5}$ (in the unit of ppb or $\mu g\ m^{-3}$) under a specific perturbation of individual meteorological parameters. Fig. 8 and Fig. 9 show the sensitivity of $O_3$ and $PM_{2.5}$ to meteorological parameters (in the unit of ppb $K^{-1}$, ppb $\%^{-1}$, $\mu g\ m^{-3}\ K^{-1,\ or}\ \mu g\ m^{-3}\ \%^{-1}$). The sensitivity is calculated by considering all the perturbations in each meteorological parameters and the corresponding concentration changes, as explained in the beginning of Section 3.3 and Figs. S8-S13. Therefore, they are not redundant.

The spatial distributions are not the same. We put Fig. 3 and Fig. 8 side by side here. The spatial distribution is similar for temperature (explained in the manuscript), but clearly they are very different for WS, PBL, CLW, and PCP. Similar finding for comparing Fig. 4 and Fig. 9.

[Figure]

Fig. 3                              Fig. 8

3. There is ambugity regarding how the authors perturbed the meteorological parameters. It is not clear to readers what processes are affected by the perturbation.
For example, when the authors perturb T in CMAQ, does absolute humidity or relative humidity change? For relationship between T, AH, and RH to be invalid, one of them has to change. But changing AH or RH can have different consequences on chemical reactions. (e.g., smaller RH can cause less aerosol water and thus impact on the SNA). Or, maybe the only thing considered is T in the chemical kinetic computation?
Similar case for PBLH. When authors claim they perturbed PBLH in CMAQ, it is most likely that they simply perturbed the PBLH parameter in the model's boundary layer mixing scheme. But a real life change in PBLH will also implicate changes in vertical profiles in T, WS, AH, etc.

Responses: In the CMAQ meteorological input files, we perturbed one parameter every time. In the case of T perturbation, we only changed T but kept all other meteorological parameters the same. This means the AH was not changes (AH is in the CMAQ meteorological input files). As the reviewer said, RH then was changed with the T perturbation. All the chemical processes that involved T and RH were then affected by the perturbation in T.

In the case for PBLH, we only changed the PBLH values in the meteorological input files, while all the other parameters were kept the same. So the vertical profiles in T, WS and AH were indeed not changed. We admit that this perturbation would not occur in the real world as the meteorological parameters in the real world are coupled together. However, we used this method to investigate each of these parameters separately.

We added a sentence of "Please note that this type of perturbations are not what happens in the real world where meteorological parameters are inter-linked." in Lines 156-158 in the revised manuscript to remind readers about this.

Minor comments:

Line 37: -1 ug m-3 %-1

Responses: Corrected.

Fig. 4: caption for y axis is wrong

Responses: Corrected.

Referee #2

I have a suggestion for the authors if they can include it will further bolster their manuscript.

The authors claim that they "isolate the effects of individual meteorological parameters" however the met parameters are inter-linked e.g. Change in temperature will result in corresponding change of PBL height, In that case how can the authors claim that they only study the isolated effect of change in temperature in this case. Is such an isolated change of a meteorological parameter possible meaning say only change of temperature without change in PBL? If not then your result from the study that "July O3 in Beijing is very sensitive to temperature and not to PBL and hence additional emission controls would be needed if temperature is predicted to increase in future" which is entirely based on presumption that either PBL or temperature will change might not give correct information. I suggest the authors can describe some uncertainties about their statement.

Responses: In the CMAQ meteorological input files, we perturbed one parameter every time. In the case of T perturbation, we only changed T but kept all other meteorological parameters (including PBLH) the same. We added a sentence of "
[revised manuscript text omitted]